# Human cell surface-AAV interactomes identify LRP6 as blood-brain barrier transcytosis receptor and immune cytokine IL3 as AAV9 binder

Timothy F. Shay [1,3] ✉, Seongmin Jang [1,3], Tyler J. Brittain [1,3], Xinhong Chen[1,3], Beth Walker[2], Claire Tebbutt[2], Yujie Fan[1], Damien A. Wolfe[1], Cynthia M. Arokiaraj [1], Erin E. Sullivan[1], Xiaozhe Ding[1], Ting-Yu Wang[1], Yaping Lei[1], Miguel R. Chuapoco[1], Tsui-Fen Chou [1] & Viviana Gradinaru [1] ✉

Adeno-associated viruses (AAVs) are foundational gene delivery tools for basic science and clinical therapeutics. However, lack of mechanistic insight, especially for engineered vectors created by directed evolution, can hamper their application. Here, we adapt an unbiased human cell microarray platform to determine the extracellular and cell surface interactomes of natural and engineered AAVs. We identify a naturally-evolved and serotype-specific interaction between the AAV9 capsid and human interleukin 3 (IL3), with possible roles in host immune modulation, as well as lab-evolved low-density lipoprotein receptor-related protein 6 (LRP6) interactions specific to engineered capsids with enhanced blood-brain barrier crossing in non-human primates after intravenous administration. The unbiased cell microarray screening approach also allows us to identify off-target tissue binding interactions of engineered brain-enriched AAV capsids that may inform vectors' peripheral organ tropism and side effects. Our cryo-electron tomography and AlphaFold modeling of capsid-interactor complexes reveal LRP6 and IL3 binding sites. These results allow confident application of engineered AAVs in diverse organisms and unlock future target-informed engineering of improved viral and non-viral vectors for non-invasive therapeutic delivery to the brain.

Adeno-associated viruses (AAVs) have become the gene delivery vector of choice at the bench and in the clinic[1,2]. Systemic administration of AAVs, such as AAV9[3–6], allows noninvasive gene delivery, particularly in large or distributed biological structures[7], but access to the brain from the periphery is restricted by the blood-brain barrier (BBB), a complex biological structure that regulates molecular access to the central nervous system (CNS)[8–10]. Systemic administration of AAVs also exposes the vectors to the host immune system[11,12] and off-target tissues[3,13]. The poor efficiency of brain targeting after systemic administration with natural serotypes often necessitates high doses that raise costs and may trigger serious adverse events[14–16]. Thus, improved vectors are needed if AAV gene therapy is to realize its full therapeutic potential.

[1]Division of Biology & Biological Engineering, California Institute of Technology, Pasadena, CA 91125, USA. [2]Charles River Laboratories, High Peak Business Park, Buxton Road, Chinley SK23 6FJ, UK. [3]These authors contributed equally: Timothy F. Shay, Seongmin Jang, Tyler J. Brittain, Xinhong Chen. ✉e-mail: tshay@caltech.edu; viviana@caltech.edu

AAV capsid engineering, particularly through directed evolution methods, has demonstrated that markedly improved efficiency in desired cell types and tissues after systemic intravenous delivery is possible[17-19]. In particular, two recently identified engineered capsids, AAV9-X1.1[20] and AAV.CAP-Mac[21], robustly transduce CNS neurons after systemic administration in macaques. As AAV capsids are applied across species, however, the enhanced tropisms of many engineered vectors can vary[20-23]. This is concerning for human clinical trials, as a capsid developed in another species that performs poorly when translated to humans may not only fail to provide therapeutic benefit but might preclude future therapies for the patient by inducing neutralizing antibodies[11].

This translational challenge of AAV engineering through directed evolution also represents an opportunity to better understand fundamental mechanisms of drug delivery to the brain. Directed evolution of engineered capsids with enhanced BBB crossing provides a platform with which researchers may survey the most efficient pathways across this barrier. While recent progress suggests that engineered AAVs may utilize diverse BBB-crossing receptors[24-27], the mechanisms of primate brain-enhanced vectors[20,21,23,28-30] remain underexplored.

To address this challenge, here we adapt Retrogenix cell microarrays[31,32] of the human membrane proteome and secretome to screen natural and engineered AAV capsid interactions with host cells. This allows us to rapidly assay more than 90% of the known human membrane proteome and secretome, including key protein classes such as receptors, transporters, and cytokines. This unbiased screen should thus include nearly all proteins exposed to an intravenously-injected AAV prior to cell internalization. Using this broad, unbiased screen, we identify several previously-unreported AAV interactions with implications for the host immune response (human interleukin 3 (IL3) binding to AAV9), enhanced BBB crossing across species (via low-density lipoprotein receptor-related protein 6 (LRP6) binding by AAV9-X1.1 and AAV.CAP-Mac), and peripheral tissue tropism (through pancreas-expressed glycoprotein 2 (GP2) binding by AAV9-X1.1 and CAP-Mac). We then characterize these capsid interactions using diverse biophysical and structural methods, including cryo-electron tomography. Finally, we functionally validate LRP6 capsid interactions in vivo in mice, in the human BBB via primary cell culture, and beyond the BBB using human pluripotent stem cell (hPSC)-derived neurons. Understanding the mechanism of action of systemic AAVs through methods such as those used here will be critical for successful vector translation and should enable design of improved vectors, as well as other therapeutic protein modalities, for specific targets[33,34].

## Results

### High-throughput screening for AAV binding partners

To screen AAV-binding proteins, we used Retrogenix cell microarrays[31,32] of the human membrane proteome and secretome, in which DNA oligonucleotides (oligos) encoding human membrane and secreted proteins are affixed at known slide locations (Fig. 1a). HEK293 cells are then grown on the slides and become individually reverse-transfected with the oligos in the corresponding pattern. AAVs that directly interact with a given protein will preferentially bind to cells expressing that protein; other slide locations define non-specific background binding. To increase confidence in binding specificity, each protein is patterned at two different locations (four locations for initial condition optimization) (Fig. 1b, c). We optimized screen conditions using previously-identified AAV and interacting protein pairs, (1) AAV9 with AAVR (KIAA0319L)[35] and (2) PHP.eB with mouse LY6A[25-27], and two different detection methods: biotin tagging and direct antibody detection (Fig. 1b, c). Biotinylated capsids were detected with fluorescent streptavidin, and unlabeled capsids were detected with an antibody whose epitope is distinct from the commonly engineered capsid variable regions (VR) IV and VIII[18,36]. As noted previously[37,38], capsid primary amine labeling levels must be tuned so

that surface modification does not interfere with key capsid binding interactions. We found that the best signal to noise ratio for duplicate spots (calculated as the average intensity of positive control spots compared to the average intensity of the rest of the slide) was achieved by directly fixing cell-bound AAVs without washing.

We proceeded with direct capsid antibody detection and validated conditions with a panel of AAV capsids, including AAV9 as well as five engineered AAV9 variants with enhanced potency in the CNS of non-human primates (NHPs) after systemic administration (Table 1, Fig. 1d)[20,21,23,28]. These include: (1) AAV.CAP-B22, a further-evolved variant of PHP.eB with a VR-IV substitution that enhances brain potency in marmosets after intravenous injection[28], (2) AAV.CAP-Mac, a VR-VIII insertion-modified AAV9 variant identified from selections in marmosets that efficiently transduces brain endothelial cells in marmosets and potently crosses the BBB to target neurons in old world monkeys[21], (3) AAV-MaCPNS1 and (4) AAV-MaCPNS2, VR-VIII insertion-modified AAV9 variants identified from selections in mice that have enhanced potency in the peripheral nervous system (PNS) as well as the CNS in NHPs[23], and (5) AAV9-X1.1, an AAV9 variant containing both VR-IV and VR-VIII modifications identified via mouse selections that potently targets brain endothelial cells in rodents and CNS neurons in macaques after intravenous injection[20].

Testing these capsids individually revealed that all except MaCPNS1 exhibited detectable AAVR binding (the exception may be due to the interfering geometry of the capsid's VR-VIII insertion[39]), whereas only CAP-B22 interacted with mouse LY6A (likely through the PHP.eB loop in VR-VIII[25-27]) (Fig. 1d and Supplementary Fig. 1). To enable higher-throughput screening, we decided to test the six capsids as a pool. Pooled testing required additional dosage optimization, first for the individual and then for the collective background binding levels of the included capsids (Supplementary Table 1). An optimal dose was determined that minimized background binding while still allowing the specific interaction of CAP-B22 with mouse LY6A to be distinguished from the five non-LY6A-interacting capsids (Fig. 1d).

After these controls, we then tested the six-capsid pool in a full screen of over 6400 proteins, including 6019 human plasma membrane proteins and secreted and cell surface-tethered proteins, as well as 397 heterodimers. This unbiased screen includes more than 90% of the human membrane proteome and secretome. We identified 22 pool hits with enhanced signal over background in each duplicate spot. To assign these hits to specific capsids in the pool, we performed follow-up deconvolution screens with each individual capsid from the pool (Fig. 1e). DNA oligos for the 22 identified hits, as well as the positive control CD86 and negative control EGFR, were affixed in duplicate locations to new slides. A negative control condition with no AAV analyte and a positive control condition with CTLA4-Fc (CD86 binder) were also included. We were able to successfully assign hits, including both membrane-localized and secreted proteins, to capsids. Some of these interactions were specific to select AAV9 variants, such as LRP6 for AAV9-X1.1 or FAM234A for AAV.CAP-B22, while others were conserved across all capsids tested, such as IL3 (Table 2).

### Validation of AAV binding interaction with interleukin-3

To validate binders from our cell microarray screen, membrane protein hits were tested for their ability to enhance AAV potency in cell culture (Supplementary Fig. 2) and secreted protein capsid-binding interactions, as well as soluble membrane protein extracellular domains, were characterized by surface plasmon resonance (SPR) (Fig. 2a, Supplementary Fig. 3). This reduced the candidate receptors to a subset of validated interactors (Table 2). We were struck by the identified interaction of AAV9 and all its recent lab-evolved derivatives with the human immunomodulatory protein IL3 because AAVs are relatively well tolerated by the immune system[40]. IL3 is produced by activated T cells as part of the inflammatory response to viral infection, triggering expansion of various immune cells and activating type I

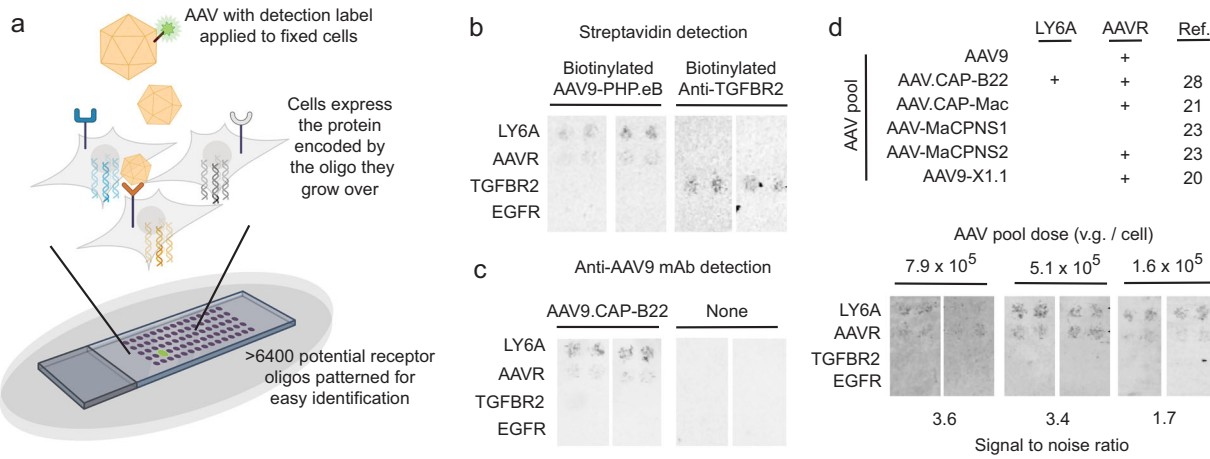

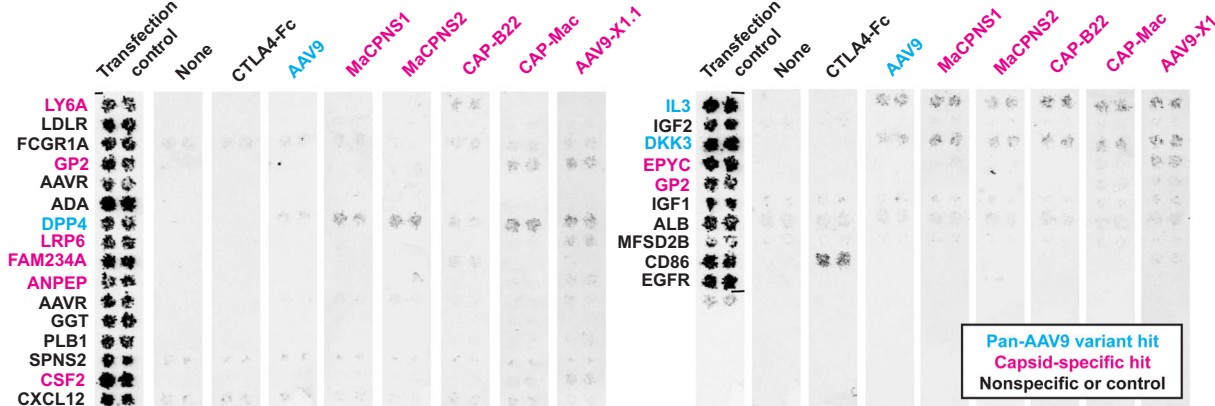

**Fig. 1 | High-throughput screen identifies AAV-binding human proteins.**
**a** Schematic of AAV cell microarray screen. DNA oligos that encode individual membrane proteins are chemically coupled to slides in a known pattern, reverse transfecting the cells that grow on them and thereby creating spots of cells over-expressing a particular, known protein. Each protein is expressed in duplicate at two different locations on the slide. When AAVs are applied to the slides, enhanced binding can be detected from duplicate cell spots overexpressing cognate AAV receptors. **b** Known AAV capsid receptor interactions, such as AAVR and LY6A with AAV-PHP.eB, were used to optimize conditions for streptavidin-based detection of biotinylated capsids with two sets of replicate spots. Anti-TGFBR2 antibody was used as a non-AAV positive control. Uncropped blots in source data. **c** AAVR and LY6A interaction with AAV9.CAP-B22 were used to optimize conditions for anti-AAV9 antibody direct detection of unmodified capsids with two sets of replicate spots. Anti-TGFBR2 antibody was used as a non-AAV control. Uncropped blots in

source data. **d** Pooled AAV capsid screening conditions were optimized by varying the concentrations of individual capsids within the pool to maximize signal to noise after direct detection with anti-AAV9 antibody, with two sets of replicate spots. v.g.: viral genomes. Uncropped blots in source data. **e** Pooled screening identified preliminary hits which were deconvoluted by individual-capsid screens, identifying previously-unreported potential capsid-binding proteins by direct detection with anti-AAV9 antibody. Transfection control condition detected fluorescent protein reverse transfected along with each receptor. None condition was treated only with anti-AAV9 antibody. Proteins in cyan were identified in all individual AAV screens, and likely represent interactions outside the engineered regions of AAV9. Proteins in magenta specifically bind to at least one engineered capsid. Uncropped blots in source data. Panel a created with BioRender.com released under a Creative Commons Attribution-NonCommercial-NoDerivs 4.0 International license.

interferon-secreting plasmacytoid dendritic cells[41]. Using SPR, we found that human IL3 binds AAV9 but not the closely related natural serotypes AAV8 and AAVrh10[42] (Fig. 2b). We then tested IL3 from different species, finding that AAV9 binds to human and macaque IL3 (83% amino acid [AA] identity) but not marmoset or mouse IL3 (69% and 27% AA identity with human, respectively) (Fig. 2c), suggesting a binding site divergence between new and old world monkeys. As interaction with human IL3 was observed in the cell microarray screen for every capsid tested (Fig. 1e), we also tested AAV9-X1.1 and observed binding with human but not mouse IL3 by SPR (Fig. 2d). The conservation of human IL3 binding across AAV9 derivatives implies that neither substitutions at VR-IV nor insertions at VR-VIII deleteriously impact this naturally evolved interaction.

To further understand the species and serotype specificity of human IL3's interaction with AAV9, we investigated the structure of the bound complex. As functional AAV ligands may have weak and

dynamic monomeric interactions[43], we leveraged avidity by flowing the 60-mer AAV9 capsid over protein A-captured dimeric IL3-Fc to ensure all biologically meaningful interactions are detected[24]. Despite this high avidity in the SPR experiment, the apparent affinity of the interaction was consistent with only a high nM interaction. Therefore, we began our structural studies by performing chemical cross-linking of IL3-bound AAV9, followed by tandem mass spectrometry (XL-MS/MS)[44]. Using bis(sulfosuccinimidyl)suberate (BS3) cross-linking agent, 2 high-confidence cross-links between the proteins were detected (Fig. 3a, Supplementary Fig. 4, and Supplementary Table 2). These cross-links place IL3's interaction site with AAV9 near the base of the threefold symmetry spike and the twofold symmetry depression, a region containing multiple residues that are specific to AAV9 compared to non-interacting serotypes (Fig. 3b). We tested chimeric AAV9 capsids with each variable region individually mutated to AAV8 amino acid identity by SPR and found that human

**Table 1 | Capsid engineering details and in vivo tropisms of vectors used in this study**

| Capsid name | AAV9 modified sequence | | Selection species | Enhanced systemic tropism in the nervous system | | | Ref. |
|---|---|---|---|---|---|---|---|
| | VR IV AA452 – 458 | VR VIII AA587 – 590 | | Adult Mouse | Adult Marmoset | Infant Macaque | |
| AAV9 (parent of vectors below) | NGSGQNQ | AQ - - - - - - - AQ | | | | | 3 |
| AAV.CAP-B10 | DGAATKN | DGTLAVPFKAQ | Mouse | CNS (neuronal) | CNS (neuronal) | | 28 |
| AAV.CAP-B22 | DGQSSKS | DGTLAVPFKAQ | Mouse | CNS | CNS (neuronal & astrocytic) | | 28 |
| AAV-MaCPNS1 | | AQPHEGSSRAQ | Mouse | PNS | CNS & PNS (neuronal & astrocytic) | CNS & PNS (neuronal & astrocytic) | 23 |
| AAV-MaCPNS2 | | AQPNASVNSAQ | Mouse | PNS | CNS & PNS (neuronal & astrocytic) | CNS & PNS (neuronal & astrocytic) | 23 |
| AAV.CAP-Mac | | AQLNTTKPIAQ | Marmoset | CNS (weak endothelial) | CNS (endothelial) | CNS (neuronal) | 21 |
| AAV9-X1.1 | DGAATKN | AQGNNTRSVAQ | Mouse | CNS (endothelial) | CNS (weak neuronal) | CNS (neuronal) | 20 |
| AAV-BI30 | | AQNNSTRGGAQ | Mouse | CNS (endothelial) | CNS (endothelial) | | 54 |

IL3 binding is most strongly determined by VR-I and VR-V (Fig. 3c and Supplementary Table 3). We used a similar chimeric strategy to substitute human IL3 residues with those from marmoset, which indicated that the N-terminus of human IL3 is critical for AAV9 binding (Supplementary Fig. 5).

Next, we turned to cryo-electron tomography (cryo-ET), which is well suited to capturing dynamic interactions and resolving low-occupancy or heterogenous ligands on the surface of AAVs[45,46]. Sub-tomogram averaging of 2661 hand-picked viral particles resulted in an initial 11.0 Å resolution map of the capsid after enforcing I1 symmetry (Supplementary Fig. 6a, b and Supplementary Table 4). While the high sigma map did not show any obvious IL3 density, adjusting the map to a low sigma suggested that such density was potentially present (Supplementary Fig. 6c, d). To bypass confounding heterogeneity in binding site occupancy in the 60mer AAV and facilitate alignment of small ligands, we performed I1 symmetry expansion and particle subtraction around the AAV trimer (Supplementary Fig. 6a, e). This overlaid each asymmetric component of the capsid on the same reference frame to extract the threefold symmetry face, generating 159,660 particles. After further 3D classification and refinement without imposing symmetry, this yielded a map of the AAV capsid threefold symmetry face that contains density protruding from a side of the threefold symmetry spike that cannot be explained by the AAV capsid alone (Fig. 3d, shown in red, and Supplementary Fig. 6f).

We performed rigid body docking of IL3 into this averaged cryo-ET density (Supplementary Fig. 7), guided by the distance constraints identified by XL-MS/MS (Supplementary Fig. 4a) and our SPR experiments showing that AAV9 VR-I and VR-V (Fig. 3c) as well as the N-terminus of human IL3 (Supplementary Fig. 5) are critical for binding. Within the sub-volume average there is density that can only be explained by one bound IL3. However, the pose suggests one IL3 would not sterically hinder another IL3 binding within the same threefold face. The IL3 binding site we identify overlaps with the known binding sites for AAVR and galactose. However, given the high number of potential binding sites on the full 60-mer capsid, the physiological relevance of this binding site overlap is uncertain.

**Validation of AAV binding interaction with LRP6**

We next assessed the validated AAV interactors for their potential to explain the enhanced brain tropisms of the engineered capsids. Sorting the screen hits by their expression level in endothelial cells of the human BBB[47] spotlighted a specific interaction of LRP6 with AAV9-X1.1 (Fig. 4a). This capsid displays enhanced brain endothelial-specific tropism in mice that shifts to enhanced neuronal tropism in macaque (Table 1)[20]. Although AAV9-X1.1 contains modifications from AAV9 at both variable regions IV and VIII (Table 1), we previously showed that the tropism of AAV9-X1.1 could be transferred to other natural serotypes such as AAV1 and AAV-DJ by transferring only the VR-VIII insertion of AAV9-X1.1[20]. By SPR, here we confirm that the X1 peptide insertion in VR-VIII endows AAV1-X1 and AAVDJ-X1, but not their unmodified parent serotypes, with LRP6 binding (Supplementary Fig. 8a). This demonstrates that the functional modularity of the X1 peptide in different AAV serotypes in vivo corresponds to LRP6-binding modularity.

LRP6 is a coreceptor of the canonical Wnt signaling pathway, with developmental and homeostatic roles in many tissues[48–50]. The high degree of LRP6 sequence conservation across species (98% and 99.5% AA identity between human LRP6 and mouse or macaque LRP6, respectively) aligns with AAV9-X1.1's enhanced tropism compared to AAV9 in rodents and primates. A similar enhancement in tropism across species is also seen for CAP-Mac[21] (Table 1), which was engineered in marmosets and has enhanced endothelial tropism compared to AAV9 in marmosets as well as enhanced neuronal tropism in macaque. Therefore, we also tested CAP-Mac by SPR for interaction with the human LRP6 extracellular domain (Fig. 4b). As with IL3, we utilized avidity to ensure that weak yet functionally important

interactions were captured. Both AAV9-X1.1 and CAP-Mac strongly bind human LRP6-Fc, unlike their parent capsid, AAV9, with a sub-nM apparent affinity.

LRP6 has many endogenous Wnt signaling partners with binding sites spanning either extracellular YWTD domains 1 and 2 (E1E2) or domains 3 and 4 (E3E4)[51]. We applied AlphaFold-Multimer[52] to build models of LRP6-AAV interaction complexes, which predicted that the X1 and CAP-Mac VR-VIII peptides bind LRP6 YWTD domain 1 (Fig. 4c). While the cooperative folding of E1 and E2 complicates testing of individual domains[53], SPR of mouse LRP6 extracellular domain fragments was consistent with the model predictions, with interaction observed for LRP6-E1E2 but not LRP6-E3E4 (Fig. 4d). We also tested AAV-BI30, another engineered capsid with specific tropism for the mouse brain endothelium[54] (Table 1), finding that it also binds to LRP6-E1E2 but not LRP6-E3E4 (Supplementary Fig. 8b). A pull-down assay confirmed that both AAV9-X1.1 and CAP-Mac bind to AAVR's PKD2 domain and the full length extracellular domain of mouse LRP6 but not

that of closely-related LRP5 (Supplementary Fig. 9a)[51]. AAV9, on the other hand, bound only to AAVR's PKD2, as reported previously[39]. Subsequent semi-quantitative pulldown experiments suggested that the strength of CAP-Mac's interaction with AAVR PKD2 is similar to that of AAV9 (Supplementary Fig. 9b). In contrast, AAV9-X1.1, like PHP.eB[39], had a slightly reduced affinity for AAVR PKD2 compared to AAV9. Interaction of both engineered capsids with galactose was unaffected in pulldown experiments (Supplementary Fig. 9c).

AAV9-X1.1 and CAP-Mac potently infect HEK293 cells, with AAV9-X1.1 having a stronger effect (Supplementary Fig. 10a). To determine if this potency is mediated by endogenous LRP6 expression in HEK293 cells, we tested the effect of LRP6 inhibitors. AAV9-X1.1 potency was markedly reduced by mesoderm development LRP chaperone (Mesd), a natural endoplasmic reticulum chaperone and exogenous extracellular inhibitor of LRP5 and LRP6[55], and sclerostin (SOST), which inhibits LRP6 through specific binding of E1E2 alone[56] (Supplementary Fig. 10). Importantly, neither Mesd nor SOST inhibited the potency of PHP.eB in LY6A-overexpressing cells. Transient overexpression of human LRP6 boosted the potency of both AAV9-X1.1 and CAP-Mac, with a stronger effect for CAP-Mac (Supplementary Fig. 10a). This effect was largely preserved with a construct excising LRP6-E3E4. As expected from our pull-down assay, transient overexpression of LRP5 did not enhance the potency of either CAP-Mac or AAV9-X1.1. Together, these results support a specific functional interaction between LRP6 and both CAP-Mac and AAV9-X1.1, although the two capsids may have different functional sensitivities to LRP6 expression level. While AAV9-X1.1 productively engages LRP6 at the lower endogenous expression levels of LRP6 on HEK293 cells, CAP-Mac shows greater potency when LRP6 is overexpressed.

Of note, in addition to the intended CNS receptors gained through AAV capsid engineering, both LRP6-binding capsids also gained interactions with the glycosylphosphatidylinositol (GPI)-linked protein

**Table 2 | Summary of identified AAV interactions**

| Vector name | Screen hits | | Validated interactions |
|---|---|---|---|
| | *Membrane* | *Secreted* | |
| AAV9 (parent of vectors below) | DPP4 | DKK3, IL3 | IL3 |
| AAV.CAP-B22 | LY6A, FAM234A, DPP4 | DKK3, IL3 | LY6A, FAM234A |
| AAV-MaCPNS1 | DPP4 | DKK3, IL3 | |
| AAV-MaCPNS2 | DPP4 | DKK3, IL3 | |
| AAV.CAP-Mac | DPP4, GP2 | DKK3, IL3 | LRP6, GP2 |
| AAV9-X1.1 | DPP4, GP2, LRP6, ANPEP | DKK3, IL3, CSF2, EPYC | LRP6, GP2 |

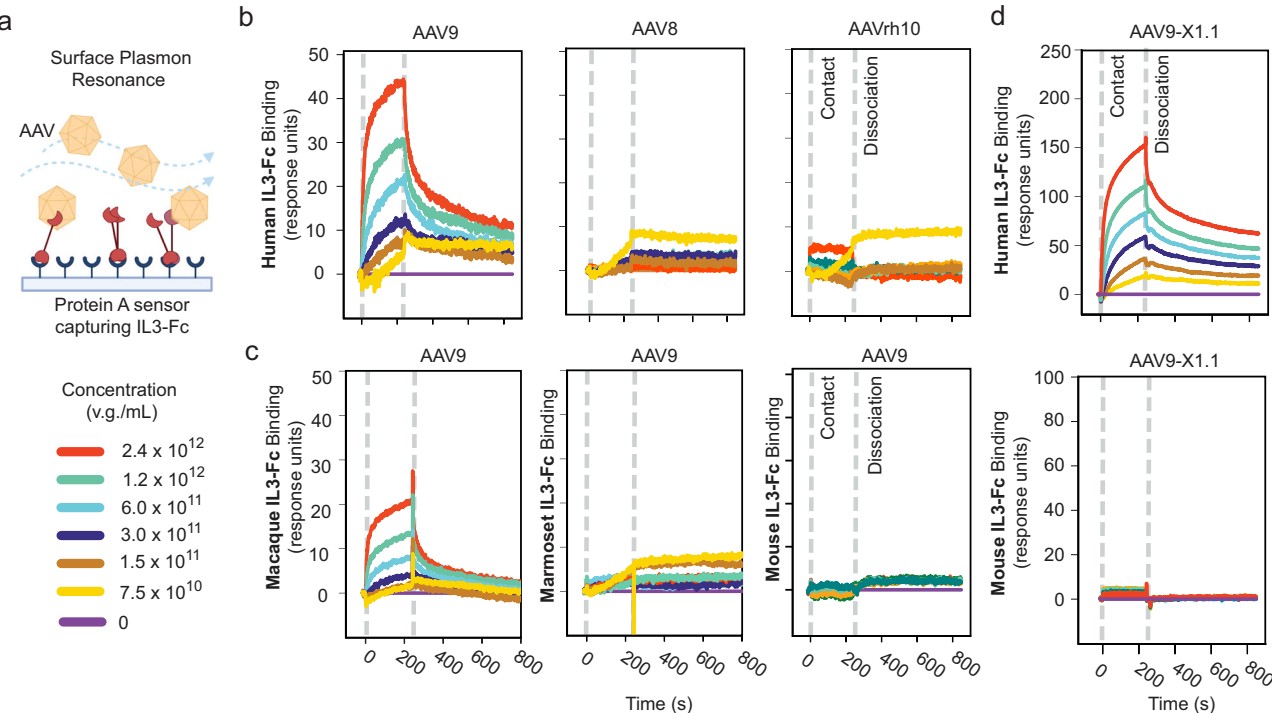

**Fig. 2 | Species and serotype-specific interaction between AAV9 and the human immunomodulatory cytokine IL3. a** Schematic of surface plasmon resonance (SPR) experiments where IL3-Fc is captured on a protein A sensor chip and AAV analyte flows over the sensor. v.g.: viral genomes. **b** SPR confirms serotype-specific interaction of AAV9 with the human immunomodulatory cytokine IL3. **c** SPR confirms AAV9 binding with macaque but not marmoset or mouse IL3. **d** SPR confirms that the VR-IV and VR-VIII modified AAV9-X1.1 capsid binds to human but not mouse IL3. Panel (**a**) created with BioRender.com released under a Creative Commons Attribution-NonCommercial-NoDerivs 4.0 International license.

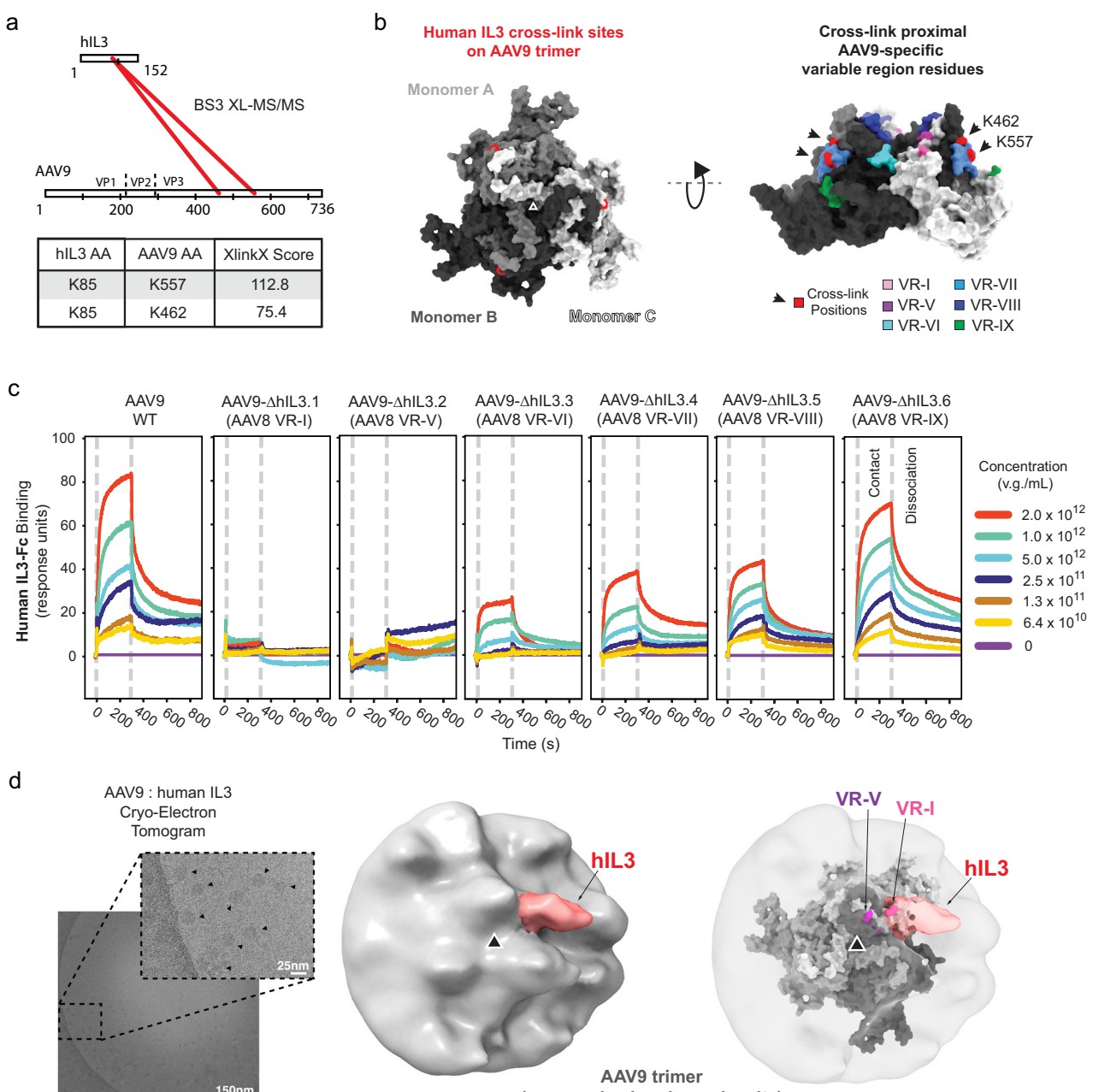

**Fig. 3 | Structural characterization of AAV9 interaction with human IL3.**
**a** Schematic depicting intermolecular cross-links between human IL3 (hIL3) and AAV9 with XlinkX scores above 40, indicating high confidence[83]. **b** Structure of AAV9 (PDB ID: 3UX1) trimer indicating human IL3 cross-linking amino acids (red) and the capsid variable regions (VR) within reach of the cross-linker (VR-I: pink, VR-V: purple, VR-VI: teal, VR-VII: cyan, VR-VIII: blue, VR-IX: green). **c** SPR of human IL3 binding for AAV9 and chimeric capsids containing AAV8 amino acid identity at the indicated variable regions. v.g.: viral genomes. **d** Left: A central slice from a cryo-

electron tomogram of AAV9 with human IL3. Arrows in magnification indicate AAV9 capsids. 46 3D tomograms, composed of 41 such images each, yielded 85,135 symmetry-expanded particles for sub-tomogram averaging. Middle: Map of AAV9 trimer bound by human IL3-Fc obtained after symmetry expansion and particle subtraction followed by multiple rounds of refinement. Human IL3-Fc density is segmented in red. Right: The same map overlaid with a model of the AAV9 trimer (PDB ID: 3UX1) highlighting the variable regions relevant for human IL3 binding (VR-I: pink, VR-V: purple).

glycoprotein 2 (GP2), which is expressed specifically in the pancreas and, in a secreted form, plays an antibacterial role in the gut[57,58] (Supplementary Fig. 2a, Supplementary Fig. 3, and Table 2). GP2 boosted the potency of both capsids in cell culture, with a stronger effect elicited by the human protein than the mouse (Supplementary Fig. 2a).

FAM234A, which bound CAP-B22 in the cell microarray screen, is found in the brain with weak expression in many neuron types[59]. Although FAM234A has been identified in disease-association studies[60], no specific molecular function has been assigned. We

found that FAM234A enhances the potency of both CAP-B22 and PHP.eB in cell culture, with the mouse protein showing a stronger effect than the human protein (Supplementary Fig. 2b). This suggests that the interaction is driven by these capsids' shared VR-VIII insertion peptide (Table 1).

## NHP brain-enhanced AAVs utilize LRP6 at the mouse BBB and in primate cell culture
Host neutralizing antibodies, developed in response to prior exposure to AAVs, complicate repeat administration with the same serotype.

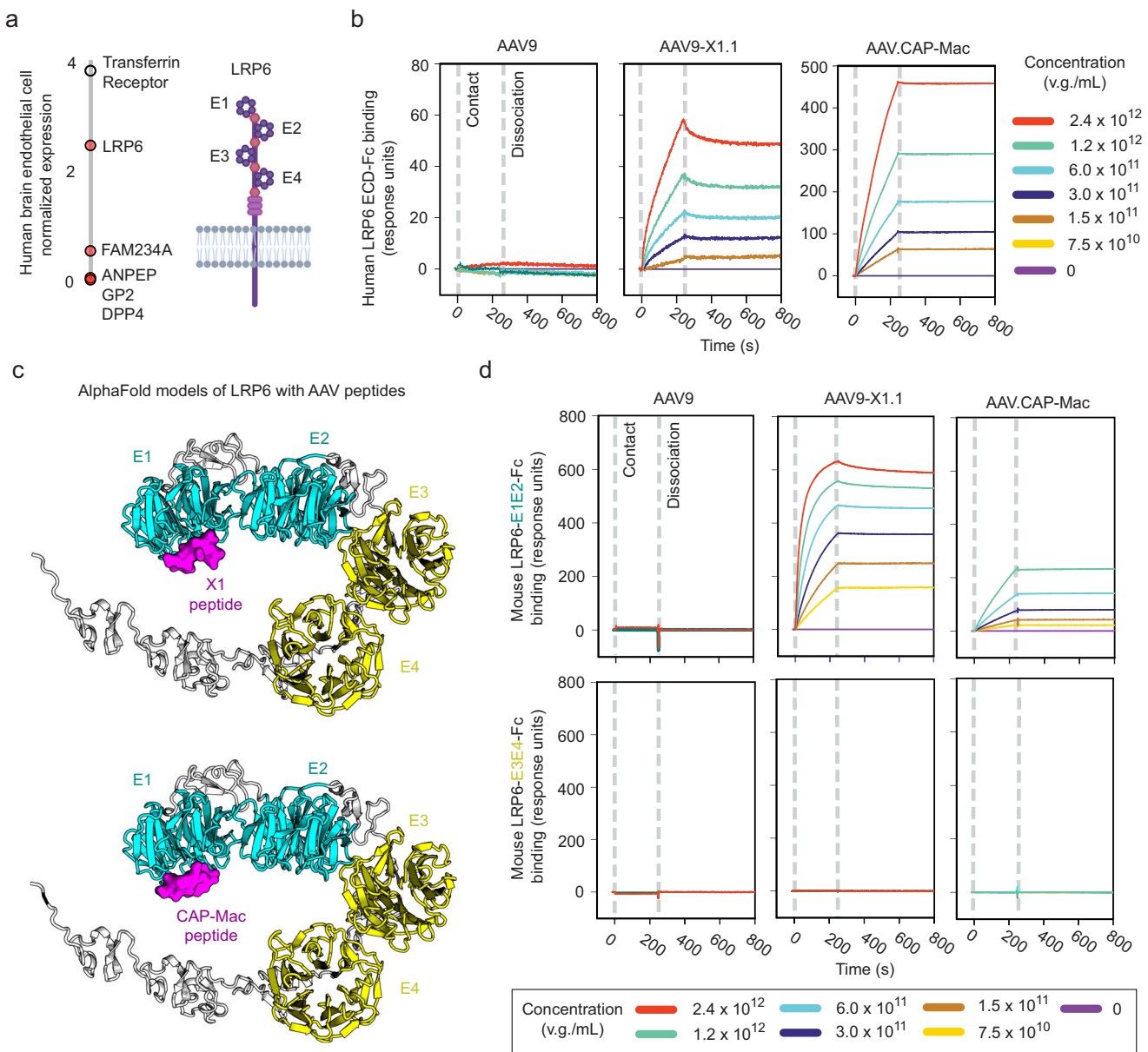

**Fig. 4 | Primate brain-enhanced AAVs gain interaction with LRP6. a** Left: Arraying AAV capsid-specific hits by human brain endothelial cell expression levels reveals highly-conserved LRP6 as a potential receptor for BBB crossing. Right: the LRP6 extracellular domain contains 4 YWTD domains (E1-E4). **b** SPR confirms that the engineered capsids AAV9-X1.1 and CAP-Mac gained direct binding interactions with human LRP6. v.g.: viral genomes. **c** Representative AlphaFold models, from 5 structural models each of X1 and CAP-Mac peptides with up to 20 recycles, predict selective interaction with human LRP6 domain E1 (E1 and E2: teal, E3 and E4: yellow, AAV peptides: purple). **d** SPR of mouse LRP6-E1E2 and LRP6-E3E4 (the minimal stable extracellular domain fragments due to cooperative folding) confirms that AAV9-X1.1 and CAP-Mac bind only to LRP6-E1E2. Panel a created with BioRender.com released under a Creative Commons Attribution-NonCommercial-NoDerivs 4.0 International license.

Serotype-switched X1 vectors, such as AAV1-X1, were shown to enable a second systemic dosing in mice previously exposed to AAV9-based vectors[20]. We leveraged this property to determine the in vivo effects of AAV9-X1.1's LRP6 interaction (Fig. 5a). Brain endothelium-targeted AAV1-X1 packaging either control mCherry or Cre recombinase was systemically administered to *Lrp6* Cre-conditional knockout mice. After three weeks, either AAV9-based PHP.eB or AAV9-X1.1 packaging eGFP was systemically delivered. Whereas PHP.eB showed characteristic strong brain transduction regardless of AAV1-X1 cargo, AAV9-X1.1 brain endothelial tropism was markedly reduced in the AAV1-X1-dosed mice with *Lrp6* knocked out in AAV1-X1 transfected cells (Fig. 5b, c), confirming the necessity of LRP6 for BBB capsid entry in vivo. The AAV9-X1.1 capsid showed enhanced potency compared to PHP.eB in

the liver, where LRP6 is also expressed[61] (Supplementary Fig. 11a). In *Lrp6* knockout conditions, decreased AAV9-X1.1 liver transduction was also observed (Fig. 5b, c).

To confirm that LRP6 interaction is mediating the previously-characterized enhanced in vivo brain potency of AAV9-X1.1 in primates[20], we tested the vector on macaque and human primary brain microvascular endothelial cells (PBMECs) in culture (Fig. 6a, b). AAV9-X1.1 was markedly more potent than its parent, AAV9, in the PBMECs from both species, and the LRP6 inhibitor Mesd selectively reduced AAV9-X1.1 potency back to AAV9 levels. A similar LRP6-dependent boost in potency for AAV1-X1 compared to AAV1 was also observed in human PBMECs (Supplementary Fig. 11b). The similarity of the responses of AAV1-X1 and AAV9-X1.1 is consistent with our SPR

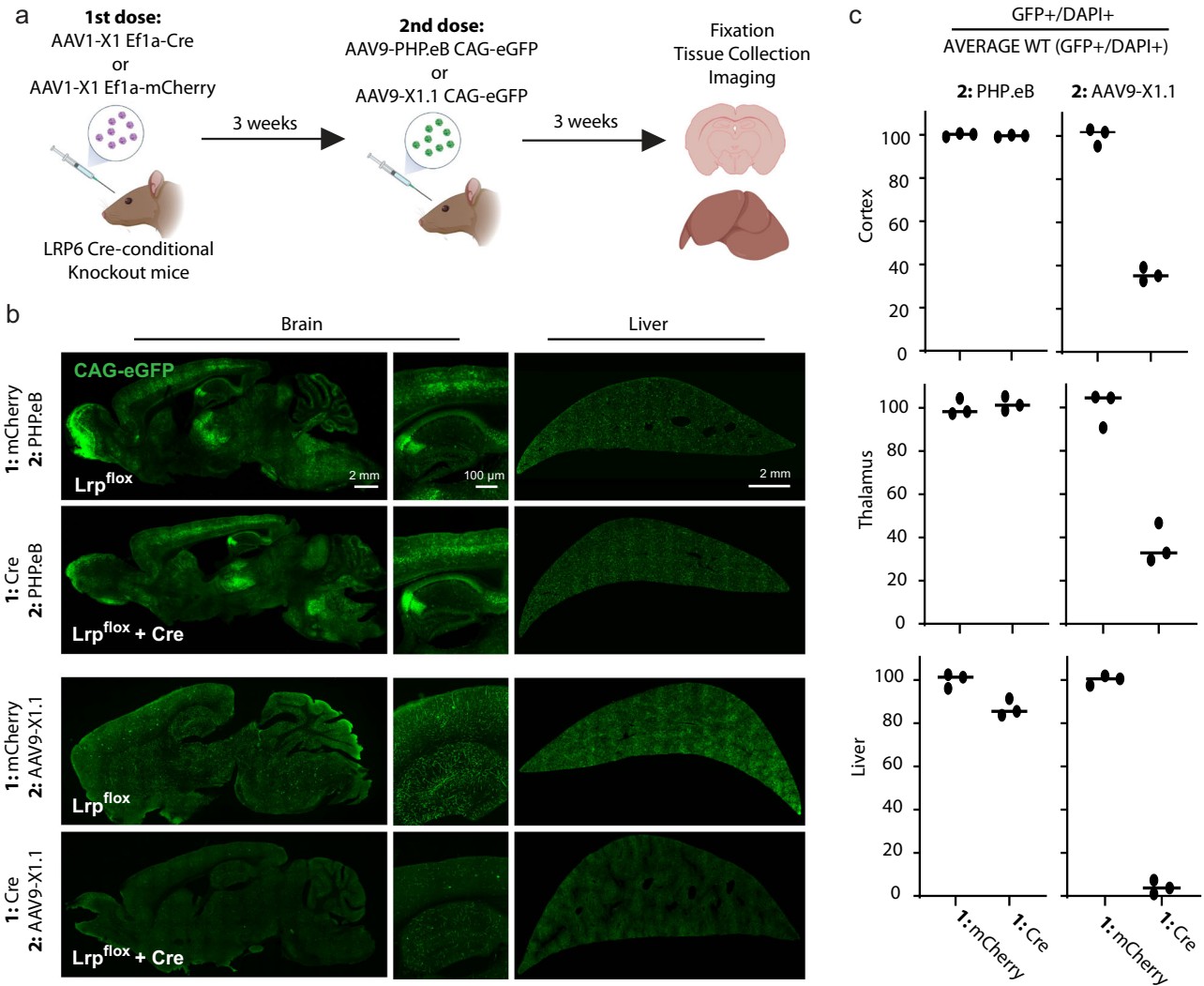

**Fig. 5 | LRP6 modulates CNS function of engineered AAVs in mice. a** Schematic of *Lrp6* conditional knockout by sequential AAV injection. Cre-conditional *Lrp6* knockout mice were systemically injected with AAV1-X1 packaging either Cre or mCherry, creating cohorts of mice that differ in their *Lrp6* expression. After allowing time for expression, these cohorts were each injected with AAV9-PHP.eB or AAV9-X1.1 packaging eGFP. By switching serotypes, neutralizing antibodies are evaded and vector dependence on *Lrp6* in vivo may be assessed. **b** Representative sagittal brain images (left) and liver images (right). Imaging parameters were

optimized independently for AAV9-X1.1 and AAV9-PHP.eB second dose conditions. **c** Quantification of AAV potency demonstrating that conditional knockout of *Lrp6* in mouse selectively and potently reduces AAV9-X1.1 brain and liver gene delivery. Data points are the average of two technical replicate sections per tissue region for each of 3 biological replicate animals, with consistent physiological regions of interest across the four experimental cohorts. Bars represent the mean value. Panel a created with BioRender.com released under a Creative Commons Attribution-NonCommercial-NoDerivs 4.0 International license.

experiments (Supplementary Fig. 8a) showing that the X1 peptide is necessary and sufficient for LRP6 interaction, and thus BBB transcytosis.

The enhanced brain potency of AAV9-X1.1 in macaque in vivo is overwhelmingly due to increased transduction of neurons[20]. As LRP6 is expressed not only on the human brain endothelium but also on neurons, we expected that the interaction of AAV9-X1.1 with LRP6 might enhance not only BBB crossing but also neuronal transduction once past the BBB. Therefore, we tested AAV9 and AAV9-X1.1 in human pluripotent stem cell (hPSC)-derived midbrain dopaminergic neurons, and found that AAV9-X1.1 strongly outperformed AAV9 (Fig. 6c). LRP6 expression was largely dependent on neuron maturity, as determined by NeuN staining, and, as expected, AAV9-X1.1 was even more potent than AAV9 in mature neurons (Fig. 6d). AAV9, on the other hand, displayed no bias across the derived cell populations. Mesd inhibition of LRP6 decreased AAV9-X1.1 potency back to AAV9 levels in mature hPSC-derived neurons, confirming a receptor-dependent effect (Fig. 6c, d).

## Discussion
Recent advances in capsid engineering have led to AAV vectors that can more efficiently cross the BBB in rodents and NHPs after systemic administration[17–19], but predictable translation and further rational design of these and other, non-viral BBB-crossing molecules is hampered by our limited understanding of transcytosis mechanisms, particularly in humans. This translational challenge is also an opportunity to better understand the biology of the BBB and AAV vectors. To date, only a few targets, such as transferrin receptor (TfR)[62], have been used for research or therapies. Here, we developed a pipeline to find cognate receptors for engineered AAVs, focusing on the human membrane proteome and secretome. Our results validate the utility of cell microarray screening to identify receptors for natural and engineered AAVs. We identify LRP6 as a previously-unreported and highly-conserved target for BBB transcytosis by AAV9-X1.1, a potent engineered capsid for primate CNS neurons after intravenous injection[20] and human IL3 as an interaction partner for AAV9. These findings offer

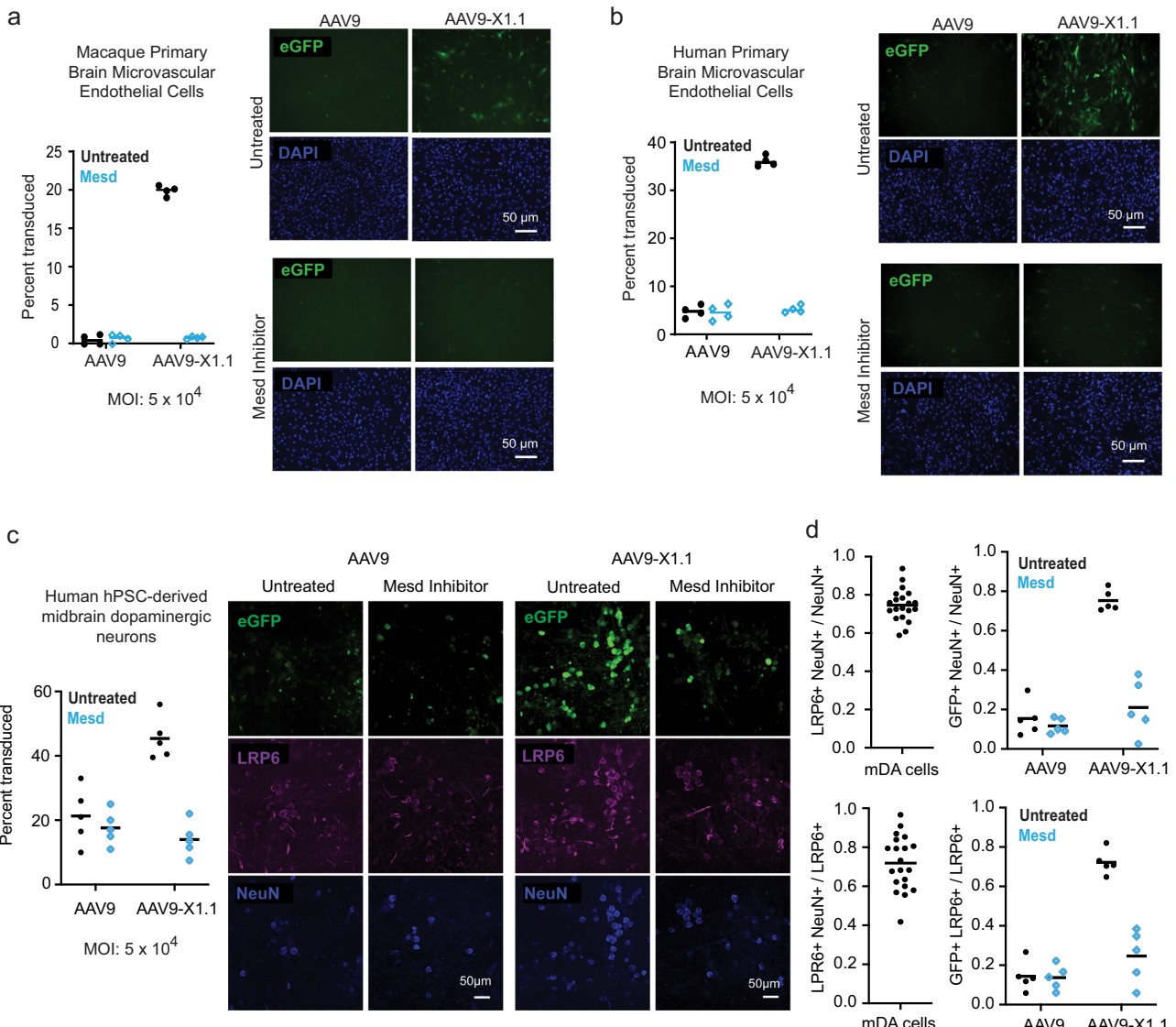

**Fig. 6 | LRP6 enhances engineered AAV potency in primate BBB and neuronal cell culture.** AAV9-X1.1 has enhanced potency in (**a**) macaque and (**b**) human primary brain microvascular endothelial cell culture (black data points), which decreases to AAV9 levels with Mesd inhibition of LRP6 (blue data points). 4 biological replicates were performed and quantified for each condition. Bars indicate the mean value. MOI: multiplicity of infection. **c** AAV9-X1.1 has enhanced potency in human pluripotent stem cell (hPSC)-derived midbrain dopaminergic neuronal culture (black data points), which decreases to AAV9 levels with Mesd inhibition of LRP6 (blue data points). 5 biological replicates were performed and quantified for each condition. **d** Quantification of LRP6 and NeuN immunohistochemistry confirms that LRP6 expression is largely restricted to mature neurons (20 biological replicates) and that the LRP6-dependent enhanced potency of AAV9-X1.1 is enriched in this population (5 biological replicates for each condition, Mesd inhibition condition in blue).

the prospect of leveraging identified receptors for targeted drug delivery across diverse therapeutic modalities, such as small molecules, antibodies, or oligonucleotides.

Delivery vector safety and immune tolerance are key considerations for AAVs moving into the clinic, as serious adverse events can occur[14–16]. Understanding the immunomodulatory potential of the IL3-AAV9 interaction we report here is therefore of high importance. Future work will have to determine whether this is a host neutralization mechanism, or a cloaking mechanism for the AAV to evade the immune system or use decoy receptors to weaken its response[63]. In addition to activated T cells, IL3 is also constitutively secreted by astrocytes in the brain to reprogram microglia, and combat Alzheimer's disease[64]. Thus, AAV9 interaction with IL3, which is shared by all AAV9-based engineered capsids with enhanced BBB crossing, may impact processes beyond immune tolerance of the vector itself in the context of healthy and diseased brains.

Engineered capsid-receptor complexes have proven challenging targets for traditional structural biology techniques. However, using a combination of biophysical techniques including cryo-ET sub-tomogram averaging with symmetry expansion, we were able to locate the human IL3 binding site on the side of AAV9's threefold symmetry spike adjacent to the capsid's twofold symmetry valley and generate an integrative binding model. Sub-tomogram averaging is well suited to the low affinity, high avidity and dynamism of receptor interactions with engineered capsids and we expect the method to have broad utility alongside single particle techniques. Importantly, the implications of the IL3 interaction cannot be readily studied in mice as AAV9, an isolate from human clinical tissue[65], binds to human and macaque (83% AA identity) but not marmoset or mouse IL3 (69% and 27% AA identity, respectively). It is possible that this species-dependent interaction could contribute to a disconnect between rodent and primate AAV safety profiles[66–69], especially in neurodegeneration contexts.

The high degree of sequence conservation of LRP6 (98% AA identity between mouse and human[61]) helps explain the broad conservation across species of enhanced tropism by AAV capsids targeting this receptor. LRP6, unlike previously-reported BBB transcytosis targets, therefore represents a receptor for NHP-enhanced AAVs with high conservation of both expression and vector-binding sites in humans. In contrast LY6A and LY6C1 are present only in rodents[70], and carbonic anhydrase IV (CA-IV) and TfR surface binding sites diverged between species, requiring transgenic animals or re-engineering of preclinical molecules prior to translation[24,71]. As a BBB target, LRP6 also benefits from lower peripheral expression levels than TfR[72], as well as more consistent human BBB expression profiles in certain disease states (e.g. Alzheimer's disease[73]). Unlike CA-IV, LY6A, and LY6C1, LRP6 is expressed not just at the BBB but also in neurons. Our experiments with hPSC-derived neurons suggest that LRP6 enhances its interacting AAV capsids' neuronal potency at multiple levels, through more efficient BBB crossing and more efficient neuronal transduction. This aligns with the enhanced neuronal potency of AAV9-X1.1 and CAP-Mac in vivo in macaques[20,21].

We could not identify a receptor for several of the engineered capsids that we screened, such as MaCPNS1 and MaCPNS2, despite their CNS potency in macaque. It is possible that this is due to a false negative (as initially observed for CAP-Mac with LRP6), reliance on a combination of receptors only screened individually here, or a receptor whose binding site is not conserved between macaque and human. The last possibility should concern those intending to translate macaque-evolved AAVs into the clinic in the absence of mechanistic knowledge.

We show that LRP6 interaction with the engineered capsids AAV9-X1.1 and CAP-Mac is layered on top of naturally-evolved AAV9 receptor interactions. Galactose interaction is unaffected by the modifications in the engineered capsids, and AAVR interaction is unaffected in CAP-Mac and only modestly weakened in AAV9-X1.1, an effect seen previously in PHP.eB[39]. We therefore expect that the in vivo behavior of AAV9-X1.1 and CAP-Mac may result from synergy of engineered and natural receptors.

That both AAV9-X1.1 and CAP-Mac also showed binding to GP2, which is expressed not in the CNS but in the pancreas and small intestine, suggests that this interaction may have piggybacked on the functional enhancement provided by LRP6-binding during directed evolution selections. This is supported by the finding that the AAVs more potently interact with human GP2 than the mouse protein that was present during the directed evolution of AAV9-X1.1. Notably, AAV9-X1.1 shows decreased transduction of small intestine and pancreas in NHP[20], which might be expected given GP2's physiological roles. GP2 is constitutively secreted by the pancreas to the gut as a host defense against bacteria[58] and is membrane-associated in small intestinal M cells, where it acts as a transcytosis receptor delivering antigens to the underlying resident dendritic cells and initiating antigen-specific mucosal immune responses[74]. These findings highlight the importance of broad, unbiased interaction screens to build full safety profiles for engineered capsids prior to clinical trials.

Surveying the diversity of mechanisms by which natural and engineered AAVs cross the BBB may also allow us to prepare defenses against future pathogens. Just as antibiotic resistance is testing our modern world, one concern is that fast-evolving pathogens will develop "BBB resistance"—the ability to access the brain and cause severe disease (as some retroviruses, including HIV-1, already do[75]). As a recent troubling example, SARS-CoV-2 capsid proteins were found in the brains of patients with long COVID, and correlated with neuropsychiatric symptoms[76]. By screening existing pathogens and their likely molecular evolutions against the growing human BBB transcytosis receptor catalog including TfR[71,77], insulin receptor[78,79], CD98hc[80,81], CA-IV[24], and LRP6 (this work), we may be able to anticipate outbreaks of pathogens with neuropsychiatric sequelae.

In summary, the present study introduces a method to efficiently screen natural AAV serotypes and engineered variants against the human proteome, and expands the limited roster of targets for enhanced BBB crossing in primates. These findings suggest strategies for successful clinical translation of engineered AAVs, provide targets for development of non-viral therapeutic modalities, and highlight latent vulnerabilities to future pathogens.

# Methods

## Animals
This research complied with all relevant ethical regulations and all mouse procedures were approved by the California Institute of Technology Institutional Animal Care and Use Committee (IACUC). Mice were group housed with 13/11 light/dark cycles at ambient temperatures of 71–75 °F and 30–70% humidity. Adult (6–8 weeks old) homozygous B6;129S-Lrp6tm1.1Vari/J mice (Jackson Labs #026267) were retro-orbitally administered $1 \times 10^{12}$ viral genomes (v.g.) per animal AAV1-X1 packaging either EF1a-mCherry or EF1a-Cre ($N = 6$ per condition). After 3 weeks, mice were re-administered $1 \times 10^{12}$ v.g. per animal PHP.eB or AAV-X1.1 packaging CAG-eGFP ($N = 3$ per condition). Mice were randomly assigned to a particular AAV condition. Experimenters were not blinded for any of the experiments performed in this study. Animals of both sexes were included but sex was not considered as a variable as AAV9-X1.1 was already shown to perform identically across sex[20].

## Viral vector production
AAVs were produced as previously described[82]. Briefly, HEK293 cells (ATCC, CRL-3216) were triple transfected with capsid, genome, and helper plasmids. Media was exchanged the next day then collected and replaced two days after. At five days post-transfection, media and cells were collected and processed for AAV purification. Cells were lysed in a high-salt solution and treated with salt-activated nuclease. Media were PEG precipitated and resuspended in salt-activated nuclease solution. Both solutions were added to iodixanol density columns, ultracentrifuged, and AAVs extracted from the 40%/60% interface. Finally, AAVs were buffer exchanged, concentrated, titered, and (for vectors destined for NHPs) assayed for endotoxin using Pierce LAL chromogenic endotoxin kit (cat# A39552).

## Retrogenix cell microarray
Retrogenix cell microarray screening was performed as previously described[31,32] with the following adaptations for AAV analytes. Pre-screen optimizations were performed on slides of HEK293 cells and cells overexpressing mouse LY6A and human AAVR (KIAA0319L), TGFBR2, and EGFR. Transfection efficiencies were validated to exceed a minimum threshold prior to analyte application. AAVs were added to fixed cells at a concentration of $6 \times 10^4$ AAV particles per HEK293 cell.

Biotinylated AAVs were created by incubation for 2 h at room temperature with 10,000-fold molar ratio of NHS-PEG4-biotin (Thermo Fisher Scientific A39259) to AAV at $1 \times 10^{13}$ v.g. per mL in PBS. Reactions were quenched with 1 M Tris, pH 8 prior to buffer exchange, concentration, and AAV re-titer. Biotinylated AAVs were detected on HEK293 cells post fixation by AF647-labeled streptavidin. Unlabeled AAVs were detected on HEK293 cells post fixation by anti-AAV9 clone HL2372 (Merck, MABF2309-100UL) at a 1:500 dilution followed by AF647-labeled anti-mIgG H + L.

To achieve a suitable signal to noise ratio, necessary for minimizing false positives and false negatives, unlabeled AAVs were screened individually and as a pool at various concentrations, using anti-AAV9 detection. The final test pool was screened against fixed HEK293 cells/slides expressing approximately 6000 human plasma membrane proteins, secreted and cell surface-tethered human secreted proteins and approximately 400 human heterodimers, each in duplicate. Hits were identified using ImageQuant as spots observed in

duplicate. Following the screen, the 22 identified hits and CD86 positive control protein were spotted on new slides for individual AAV testing in a deconvolution screen. A negative control condition with no analyte and a positive control condition with CTLA4-Fc (to interact with CD86) were also included.

## Protein preparation

Lyophilized mouse LRP6 (AA 20-1366) with a 6xHis tag, N-terminal (E1E2) and C-terminal (E3E4) fragments of mouse LRP6 extracellular domain (N-half: AA 20-628, C-half: AA 629-1244) tagged with mouse IgG$_{2a}$ Fc, full-length human LRP6 (AA 20-1368) tagged with human IgG$_1$ Fc, LRP5 (AA1-1383) with a 6xHis tag, and SOST protein were purchased from Bio-Techne (cat# 2960-LR-025, 9950-LR-050, 9954-LR-050, 1505-LR-025, 7344-LR-025/CF, 1406-ST, respectively). Mesd protein was purchased from SinoBiological (cat# 10949-H08H). All proteins were reconstituted in Dulbecco's phosphate-buffered saline (DPBS, Gibco™) at desired concentrations before use.

Human, macaque, marmoset, mouse and chimeric interleukin 3 (human IL3: AA 1-152, macaque IL3: AA 1-144, marmoset IL3: AA 1-143, mouse IL3: AA 1-166, chimeras: AA1-152) triple tagged with human IgG$_1$ Fc-Myc-6xHis, human and mouse GP2 (hGP2: AA 1-518, mGP2: AA 1-515) triple tagged with human IgG$_1$ Fc-Myc-6xHis, and human and mouse DKK3 (hDKK3: AA 1-350, mDKK3: AA 1-349) triple tagged with human IgG$_1$ Fc-Myc-6xHis were transfected into Expi293F™ cells (Thermo Fisher, Cat# A14527) at a density of $3 \times 10^6$ viable cells/mL using ExpiFectamine™ (Thermo Fisher) according to the manufacturer's protocol, and secreted proteins in media were harvested after 120 h and cleared using a 0.45-μm PVDF vacuum filter (Sigma Millipore). Each His-tagged protein in media was captured with Ni-NTA resin (Qiagen) and eluted with DPBS containing 150 mM imidazole.

Human Adeno-Associated Virus Receptor (AAVR) PKD2 domain (AA 401-498) tagged with 6xHis was purified as described previously[39]. Briefly, PKD2 was expressed in BL21(DE3)-RIPL E. coli. Cells were lysed by sonication, and the insoluble fraction was cleared by centrifugation. Cleared lysate was applied to a Ni-NTA column (Qiagen) and eluted using DPBS containing 250 mM imidazole.

## Surface plasmon resonance (SPR)

A Sierra SPR-32 instrument (Bruker) loaded with a protein A sensor chip was used. Fc-fusion proteins in HBS-EP+ buffer (GE Healthcare) were immobilized at a capture level of 600-800 response units (RU) for Figs. 2b–d, 3c, 4d, Supplementary Fig. 3, Supplementary Fig. 5b and Supplementary Fig. 8, and 1200-1500 RU for Fig. 4b. AAVs were injected at a flow rate of 10 μL per min for 240 seconds followed by a 600 second dissociation. AAV concentrations began at $2.4 \times 10^{12}$ v.g. per mL and proceeded at twofold dilution intervals. A regeneration step with 10 mM glycine pH 1.5 was performed between each cycle. All kinetic data were double reference-subtracted.

## Pull-down assays

The AAVR PKD2 pull-down assay was performed as described previously[39]. Briefly, prey AAVs were mixed with His-tagged bait protein and Ni-NTA resin in a binding buffer of DPBS containing 20 mM imidazole for 1 h at 4 °C on an orbital mixer. Resin was then collected in a spin column, washed twice with 10 column volumes of binding buffer and eluted in 45 μL of DPBS containing 150 mM imidazole. Eluate was analyzed by Western blot using anti-VP1/VP2/VP3 (ARP, cat# 03-61058) and anti-6xHis (Abcam, cat# ab18184) antibodies.

For the D-galactose pull-down assay, D-galactose agarose resin (Thermo Fisher, cat# 20372) was prepared by mixing with Ni-NTA resin at ratios of 1:1, 1:2, and 0. Prey AAVs in DPBS were then mixed with D-galactose resin for 1 h at 4 °C on an orbital shaker. Resin was then collected in a spin column, and processed as described above for AAVR PKD2 pull-downs.

## HEK293 cell culture potency assay

HEK293T cells (ATCC, CRL-3216) in Dulbecco's Modified Eagle Medium (DMEM) containing 5% fetal bovine serum (FBS), 1% non-essential amino acids (NEAA), and 100 U per mL penicillin-streptomycin were cultured in 6-well plates at 37 °C in 5% CO$_2$. At 80% confluency, cells were transiently transfected with 2.53 μg plasmid DNA encoding a membrane protein hit from the Retrogenix cell microarray screen. Cells were transferred to 96-well plates at 20% confluency and maintained in FluoroBrite™ DMEM supplemented with 0.5% FBS, 1% NEAA, 100 U per mL penicillin-streptomycin, 1x GlutaMAX, and 15 μM HEPES. Plates were imaged 24 h after application of AAV on a Keyence BZ-X700 (4x objective). For experiments with protein inhibitors, Mesd (26 μg/ml) and SOST (0.2 μg/ml) were added 4 h prior to AAV addition. NucBlue™ Live ReadyProbes™ reagent (Hoechst 33342) was added to each well to aid autofocusing. Image quantification was performed as described previously[24], using our custom Python image processing pipeline, available at https://github.com/GradinaruLab/in-vitro-transduction-assay.

## Cross-linking mass spectrometry (MS)

The cross-linking procedure was modified from the manufacturer's instructions (Thermo Fisher). In brief, purified AAV9 was complexed with purified human IL3 at a 1:2 ratio of 300 μM AAV & 600 μM hIL3, respectively. Bis(sulfosuccinimidyl)suberate (BS3) (Thermo Fisher) was added to a final concentration of 3 mM and incubated at room temperature for 1 h. After incubation, Tris buffer was added to a final concentration of 20 mM to quench the reaction. The samples were then run on an SDS poly-acrylamide gel and stained with Coomassie blue. Bands corresponding to cross-linked protein were cut out of the gel and further processed for mass spectrometry analysis with $n = 1$ biological replicate and $n = 1$ technical replicate.

The cut-out gel bands were washed with 50 mM NH$_4$HCO$_3$ in 50% acetonitrile and dehydrated with 100% acetonitrile before drying. The dried sample was reduced with 10 mM DTT and then alkylated with 100 mM chloroacetamide. The sample was then dehydrated with acetonitrile and dried before overnight digestion with a 20 ng per μL solution of trypsin in 50 mM NH$_4$HCO$_3$. Digestion was arrested with 5 μL of 5% formic acid. The sample was then centrifuged and the supernatant containing digested peptides was collected. Digested peptides were then desalted using ZipTip according to the manufacturers protocol (Millipore). Desalted peptides were then eluted, dried, and then suspended in LC-MS-grade water containing 0.2% formic acid and 2% acetonitrile for LC-MS/MS analysis. LC-MS/MS analysis was performed with an EASY-nLC 1200 (Thermo Fisher) coupled to a Q Exactive HF hybrid quadrupole-Orbitrap mass spectrometer (Thermo Fisher). Peptides were separated on an Aurora UHPLC Column (25 cm × 75 μm, 1.7 μm C18, AUR3-25075C18, Ion Opticks) with a flow rate of 0.35 μL/min for a total duration of 43 min and ionized at 2.2 kV in the positive ion mode. The gradient was composed of 6% solvent B (2 min), 6–25% B (20.5 min), 25-40% B (7.5 min), and 40–98% B (13 min); solvent A: 2% acetonitrile and 0.2% formic acid in water; solvent B: 80% acetonitrile and 0.2% formic acid. MS1 scans were acquired at the resolution of 60,000 from 375 to 1500 m/z, AGC target 3e6, and maximum injection time 15 ms. The 12 most abundant ions in MS1 scans were selected for fragmentation via higher-energy collisional dissociation (HCD) with a normalized collision energy (NCE) of 28. MS2 scans were acquired at a resolution of 30,000, AGC target 1e5, maximum injection time 60 ms. Dynamic exclusion was set to 30 s and ions with charge +1, +7, +8 and >+8 were excluded. The temperature of the ion transfer tube was 275 °C and the S-lens RF level was set to 60.

For cross-link identification, MS2 fragmentation spectra were searched and analyzed using Sequest and XlinkX nodes bundled into Proteome Discoverer (version 2.5, Thermo Scientific) against in silico tryptic digested protein sequences including AAV9 capsid protein VP1

and human IL3 retrieved from UniProt (Q6JC40 and Q6NZ78, respectively). The maximum missed cleavages was set to 2. The maximum parental mass error was set to 10 ppm, and the MS2 mass tolerance was set to 0.05 Da. For BS3 cross-links, variable cross-link modifications were set as DSS (K and protein N-terminus, +138.068 Da) and the dynamic modifications were set as DSS hydrolyzed on lysine (K, +156.079 Da), oxidation on methionine (M, +15.995 Da), protein N-terminal Met-loss (−131.040 Da) and protein N-terminal acetylation (+42.011 Da). Carbamidomethylation on cysteine (C, +57.021 Da) was set as a fixed modification. The false discovery rate (FDR) for cross-linked peptide validation was set to 0.01 using the XlinkX/PD Validator Node and cross-links with XlinkX score[83] greater than 40 were reported here. Mass spectrometry proteomics data have been deposited to the ProteomeXchange Consortium via the PRIDE[84] partner repository with the dataset identifier PXD045380. Identified cross-links were visualized using xiSPEC[85].

### Cryo-electron tomography and data processing

Purified human IL3 and AAV9 were mixed in a 2:1 ratio at a final concentration of 200 µM and 100 µM, respectively. The complexed sample was applied to a Quantifoil R 1.2/1.3 300 mesh Cu grid which had been glow-discharged (Pelco EasiGlow, 10 mA, 1 min). Samples were plunge-frozen using a Mark IV Vitrobot (FEI, now Thermo Fisher) (23 °C, 100% humidity, blot force 1, blot time 4 s). 51 tilt series were collected on a 300 kV Titan Krios microscope (Thermo Fisher) equipped with a K3 6k x 4k direct electron detector (Gatan). Data were collected using SerialEM software[86] with a pixel size of 2.65 Å (x33,000 magnification) at a 3 µM defocus using a dose-symmetric tilt scheme from −60° to 60° with 2° increments with total electron dose limited to 60 electron/Å$^2$.

Raw movies were binned by 2 and gain- and motion-corrected in Warp[87]. Assembled tilt-series were exported into IMOD[88] and aligned using patch tracking. Aligned tilt-series were then imported back into Warp for CTF correction and full tomogram reconstruction at a pixel size of 10 Å. Tomograms were then imported into Dynamo for manual particle selection, which resulted in 2661 particles. Particle positions were imported back into Warp and used to extract sub-volumes at a pixel size of 10 Å and a 40-pixel box size.

Sub-tomogram averaging was performed in Relion (version 3.1.3)[89] as outlined in Supplementary Fig. 6. Briefly, sub-tomograms were iteratively refined initially enforcing I1 symmetry, then reconstructed at a pixel size of 5 Å/px. After reaching an 11.0 Å resolution map, particles were symmetry expanded and local sub-volumes were created with particle subtraction, centering on one trimer of the AAV9 capsid at a pixel size of 5 Å. This was iteratively refined, and 3D classification was used to remove trimers without bound IL3. This subset of particles was further refined to produce a trimer map at 10.0 Å resolution, which was used to build a model of the human IL3-AAV9 interaction through manual rigid-body docking of the human IL3 structure (PDB ID: 5UV8)[90] onto the AAV9 trimer (PDB ID: 3UX1)[91]. Because our aim was simply to determine the IL3 binding site, we did not exclude potential duplicate particles; therefore, any resolution measurement could be an overestimate[92] and we do not report it for the resulting trimer face map. Structure visualizations in figures were prepared using UCSF ChimeraX (version 1.6.1)[93].

### Primary cell culture potency assay

Human brain microvascular endothelial cells (ScienCell Research Laboratories, cat# 1000) and cynomolgus macaque primary brain microvascular endothelial cells (CellBiologics, cat# MK-6023) were cultured as per the vendors' instructions. The cell cultures were treated with AAVs packaging single-stranded CAG-eGFP genome at a multiplicity of infection (MOI) of $5 \times 10^4$ per well (4 wells per vector). The fluorescence expression of the culture was inspected and quantified one day after the infection procedure.

### Human pluripotent stem cell culture and neuron differentiation

Human pluripotent stem cells (hPSCs) (CSES07 obtained from Cedars-Sinai Medical Center, NIH approval number: NIHhESC-11-0108) were maintained and cultured as described previously[94]. Briefly, hPSCs were cultured on 10-cm dishes coated with Vitronectin (Thermo Fisher, A14700) and maintained in E8 medium (Thermo Fisher, A1517001). The cells were split every 3–5 days at 70-85% confluence. At 80% confluence, hPSCs were dissociated to single cells and replated on Geltrex-coated (Thermo Fisher, A1413201) plates. hPSCs were first differentiated into neuron progenitors following a bi-phasic WNT activation protocol and then further differentiated into midbrain dopaminergic (mDA) neurons in neuron maturation medium, as described previously[95].

### Viral infection of hPSC-derived mDA neurons

At day 50 of differentiation, mDA neurons were treated with single-stranded AAV9 or AAV9-X1.1 packaging CAG-eGFP at an MOI of $5 \times 10^4$ per well (5 wells per vector). For experiments with LRP6 inhibitor, Mesd (25 µg/ml) was added 4 h prior to AAV. Virus and inhibitor were removed 12 h post-infection through media exchange. The fluorescence expression of the culture was inspected and quantified 14 days after the infection procedure. Neurons were stained with anti-NeuN clone 1B7 (Abcam, ab104224, diluted 1:500) or anti-LRP6 (Thermo-Fisher, PA5-89161, diluted 1:200). Image quantification was performed in Imaris using spot detection and colocalization analysis.

### *Lrp6* conditional knockout tissue preparation and imaging

Mice were anesthetized with Euthasol (pentobarbital sodium and phenytoin sodium solution, Virbac AH) and transcardially perfused with approximately 50 mL of 0.1 M PBS, pH 7.4 followed by an equal volume of 4% paraformaldehyde (PFA) in 0.1 M PBS. Collected organs were post-fixed in 4% PFA overnight at 4 °C, washed, and stored in 0.1 M PBS with 0.05% sodium azide at 4 °C. A Leica VT1200 vibratome was used to prepare 100 µm brain sections that were imaged on a Zeiss LSM 880 confocal microscope using a Plan-Apochromat 10 × 0.45 M27 (working distance, 2.0 mm) objective. Images were analyzed in Zen Black 2.3 SP1 (Zeiss) and ImageJ.

### AlphaFold structure modeling

The complex structures of the LRP6 extracellular domain and AAV-X1 or AAV.CAP-Mac VR-VIII peptide were modeled using a cloud-based implementation of AlphaFold-Multimer-v3[52] provided in ColabFold v2.3.5[96]. The input comprised two sequences: surface-exposed residues in VR-VIII of AAV-X1 (587-AQGNNTRSVAQAQTG-594) or AAV-CAP-Mac (587-AQLNTTKPIAQAQTG-594) and the extracellular domain of human LRP6 (UniProt entry O75581, residues 20-1370). We ran the Google Colaboratory notebook using an A100 SXM4 40GB GPU. Five structure models were produced using a protocol with up to 20 recycles, and MSA generated with MMseqs2 (UniRef+Environmental)[97] and templates from PDB70. The structure models were ranked using a weighted combination of pTM and iPTM scores as described in[52]. All structure visualizations in figures were prepared using PyMOL (www.pymol.org).

### Statistics & reproducibility

No statistical methods were performed, including to predetermine sample size, and no data were excluded. Experiments were not randomized and the investigators were not blinded to allocation during experiments and outcome assessment. Each in vitro experiment was evaluated in multiple repeats. Details are provided in relevant figure legends and methods sections. Animal experiments used three animals. All attempts at replication were successful.

### Reporting summary

Further information on research design is available in the Nature Portfolio Reporting Summary linked to this article.

## Data availability

Cryo-ET data has been deposited to the EMDB with accession codes EMD-42063 (I1 map) [https://www.ebi.ac.uk/pdbe/entry/emdb/EMD-42063] and EMD-41918 (trimer face) [https://www.ebi.ac.uk/pdbe/entry/emdb/EMD-41918]. Cross-linking mass spectrometry data has been deposited to the ProteomeXchange Consortium via the PRIDE[84] partner repository with the dataset identifier PXD045380. All other data supporting the findings of this study are provided as source data files. Previously published data used in the present study include: IL3 structure, PDB ID: 5UV8; AAV9 structure, PDB ID: 3UX1. Source data are provided with this paper.

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

## Acknowledgements

We thank Catherine Oikonomou for help with manuscript editing. We thank Helen McBride for assistance in establishing collaborations with Charles River Laboratories, Brad Gartland and Lynsey Chatham of Charles River Laboratories for technical assistance, and Máté Borsos for assistance breeding *Lrp6* conditional KO mice and providing AAV8 and AAVrh10. We thank Nathan Appling for helpful discussion. Cryo-electron microscopy was performed in the Beckman Institute Resource Center for Transmission Electron Microscopy at Caltech. Cross-linking mass spectrometry was performed in the Beckman Institute Proteome Exploration Laboratory. Surface plasmon resonance was performed in the Beckman Institute Protein Expression Center. Figures were created using imagery from BioRender. This project was supported by the Center for Molecular and Cellular Neuroscience in the Tianqiao and Chrissy Chen Institute for Neuroscience at Caltech (to V.G.), the Beckman Institute CLOVER Center (to T.F.S. and V.G.), NIH PIONEER DP1NS111369 (to V.G.), and NIH BRAIN Initiative Armamentarium UF1MH128336 (to V.G. and T.F.S.).

## Author contributions

T.F.S., S.J., and V.G. conceived the project. T.F.S., S.J., and V.G. wrote the manuscript and prepared figures with input from all authors. T.F.S., X.C., E.E.S., Y.L., S.J., and M.R.C. produced AAVs. B.W. and C.T. performed cell microarray screening. S.J. produced recombinant receptor protein and performed pull-down assays, T.J.B., T.Y.W., and T.F.C. performed cross-linking mass spectrometry experiments. C.M.A. and E.E.S performed cell culture potency assays. T.F.S. and S.J. performed SPR experiments. X.D. performed AlphaFold modeling. X.C. and D.A.W. performed mouse experiments. X.C. performed primary cell culture experiments. T.J.B. collected and T.J.B. and S.J. analyzed cryo-electron tomography data. Y.F. performed hPSC cell culture experiments. T.F.S and V.G. supervised and funded the project.

## Competing interests

The California Institute of Technology has a patent pending for the delivery methods identified in this manuscript, with T.F.S., X.C., S.J., and V.G. listed as inventors (PCT Patent Application No: PCT/US2024/0139329) and a provisional patent for the sequences described in this manuscript, with S.J., T.J.B., T.F.S., and V.G listed as inventors. V.G. is a co-founder and board of directors member of Capsida Therapeutics, a fully integrated AAV engineering and gene therapy company. T.F.S and V.G. are co-founders and X.C. and X.D. are co-founders and employees of Receptive Biotherapeutics. B.W. and C.T. are employees of Charles River Laboratories. The remaining authors declare no competing interests.
