## [Peer Review File · Nature Communications]

Human cell surface-AAV interactomes identify LRP6 as blood-brain barrier transcytosis receptor and immune cytokine IL3 as AAV9 binderEditorial Note: This manuscript has been previously reviewed at another journal. This document only contains reviewer comments and rebuttal letters for versions considered at Nature Communications.

REVIEWERS' COMMENTS

Reviewer #1 (Remarks to the Author):

I appreciate that the authors have addressed my comments and additionally highlighted the IL3 part of their manuscript including the modulation of AAV-IL3 interaction.

To my understanding, they argue that because of the LRP6 conservation, further engineering is unnecessary, which seems to somewhat contradict the original statements in this manuscript (unless the authors were specifically referring to IL3 all along and not LPR6?)?

Generally, I believe this work would benefit from splitting it up over two separate papers since, along the lines of reviewer #2, I struggle to see how and why both LPR6 and IL3 need to be presented in the same manuscript?

Reviewer #1 (Remarks to the Author):

I appreciate that the authors have addressed my comments and additionally highlighted the IL3 part of their manuscript including the modulation of AAV-IL3 interaction.

We thank the reviewer for their feedback.

To my understanding, they argue that because of the LRP6 conservation, further engineering is unnecessary, which seems to somewhat contradict the original statements in this manuscript (unless the authors were specifically referring to IL3 all along and not LPR6?)?

Our Abstract and Introduction refer to the benefits of knowing vector mechanisms: confident translation across species and opportunity for rational vector optimization. Identification of LRP6 provides the former and we demonstrate the latter here with IL3. We also provide citations for prior examples of vector mechanism informing vector optimization. Our statements are also forward-looking summaries that apply both to the receptors identified here and our developed method, which is now available for broad application.

Generally, I believe this work would benefit from splitting it up over two separate papers since, along the lines of reviewer #2, I struggle to see how and why both LPR6 and IL3 need to be presented in the same manuscript?

We respectfully disagree. Inclusion of all key capsid interactions identified from our newly adapted cell microarray screen in one source affirms the utility of this method and, as indicated in response to the reviewer question above, demonstrates distinct benefits of such identified mechanisms.